# Mechanics and Natural Philosophy in History

**Danilo Capecchi** [1] and **Giuseppe Ruta** [2,3,*]

1   Department of Structural and Geotechnical Engineering, Faculty of Architecture, University "La Sapienza", I-00197 Rome, Italy; danilo.capecchi@uniroma1.it
2   Department of Structural and Geotechnical Engineering, Faculty of Civil and Industrial Engineering, University "La Sapienza", I-00184 Rome, Italy
3   National Group for Mathematical Physics, I-00185 Rome, Italy
*   Correspondence: giuseppe.ruta@uniroma1.it; Tel.: +39-(0)-644585085

**Definition:** This entry presents a historical view of the meaning attributed to the terms *mechanics* and *natural philosophy*, from a hint to ancient Greece, the Middle Ages, and the Renaissance to a special focus on the 18th Century, which represents a turning point for the development of modern physics and science in general. Since we are not concerned with the summation of the histories of natural philosophy and mechanics, but only with their interrelations, this makes a detailed description of the two disciplines unnecessary.

**Keywords:** mechanics; mathematics; natural philosophy; Aristotle; Euclid

## 1. Introduction

Mechanics and natural philosophy are two locutions with completely different meanings today. Mechanics is normally understood as the science that studies the equilibrium and motion of bodies and is a highly formalized discipline; with this meaning, it is also labeled as "classical" in academic circles, so as to juxtapose it with relativistic and quantum mechanics. In the engineering world, the adjective "applied" is often juxtaposed with the noun "mechanics" to denote a discipline devoted to the investigation of technical applications of kinematics, dynamics, and statics of rigid and deformable bodies, and their encompassing fluids and gases. In this sense, mechanics is linked with the history of technicians and technology, starting from the ancient architects and manufacturers who also wrote about their techniques—one remarkable instance being the Latin architect Vitruvius.

Natural philosophy is a less common locution, and it is not possible, within the limits of this entry's contribution, to describe all its meanings with enough detail; with some gross simplification, it can be said that, for some historians of science, it is an extinct discipline that took care of searching for explanations per causas of natural phenomena; for others, it is the name of a new branch of philosophy that studies nature as a whole, including mental states [1]. It is worth mentioning the existence of an important association of scientists named the *Society for Natural Philosophy*, founded by the mathematician Clifford Ambrose Truesdell in 1963, which "nourishes specific research aimed at the unity of mathematical and physical science" (from the by-laws of the society itself). The members of this society are scholars of exact sciences: mathematics, physics, biology, chemistry, meteorology, and so on. The name of the society is due, on the one hand, to historical continuity, since in English-speaking countries, the terms "physics" and "natural philosophy" were long exchangeable; on the other hand, it is meant to stress that the problems studied by the ancient natural philosophy are now the fields of investigation of exact sciences.

As a matter of fact, historically, the locutions *mechanics* and *natural philosophy* did not mean unrelated branches of knowledge. The following quotation gives a very good idea of how the two disciplines have been related in the past.

Natural philosophy encompassed all natural phenomena of the physical world. It sought to discover the physical causes of all natural effects and was little concerned with mathematics. By contrast, the exact mathematical sciences—such as astronomy, optics, and mechanics—were narrowly confined to various computations that did not involve physical causes. Natural philosophy and the exact sciences functioned independently of each other. Although this began slowly to change in the late Middle Ages, a much more thoroughgoing union of natural philosophy and mathematics occurred in the seventeenth century and thereby made the Scientific Revolution possible. The title of Isaac Newton's great work, The Mathematical Principles of Natural Philosophy, perfectly reflects the new relationship. Natural philosophy became the "Great Mother of the Sciences", which by the nineteenth century had nourished the manifold chemical, physical, and biological sciences to maturity, thus enabling them to leave the "Great Mother" and emerge as the multiplicity of independent sciences we know today [2]. (Backcover).

The quotation compares science and natural philosophy, but it is equally valid if science is intended to be limited to mechanics, and natural philosophy to the study of motion.

Before going further, as many of the terms in use in the past and still in use today have changed their meaning, a clarification is needed:

- *Traditional natural philosophers:* people trained in the investigation of nature by using the concepts of matter, causation, ethics. Examples are Aristotle, Plato, Descartes, and Leibniz, but not Newton and many of his successors.
- *Mathematicians:* people trained in theoretical mathematics and in practical activities.

Of course, one can envision a spectrum of intermediate figures ranging between canonical philosophers and mathematicians.

## 2. Mechanics and Natural Philosophy in Ancient Greece

From ancient Greece to the Renaissance, the term *mechanics* meant one of the so-called mixed mathematics (this is a Renaissance term). In particular, it was the one that dealt with the raising of heavy bodies by means of suitable machines and concentrated on the study of these machines—all of them being reducible to the lever— but not to the inclined plane, of which the mathematical relationship between the dragged weight and the necessary dragging force was not even known.

In ancient Greece, mechanics, like geometry, underwent a process of formalization that made it a rigorous discipline that could seem purely rational [3]. The point of greatest perfection came with Archimedes and his *Equilibrium of the planes*, where an axiomatic theory of the balance, or lever, was formulated based on a few postulates (7). In order to give an idea of the level of refinement of the approach, still unsurpassed today, hereafter, the statement of some postulates and the proof of a proposition are reported. The postulates are:

1. Equal weights at equal distances are in equilibrium, and equal weights at unequal distances are not in equilibrium, but incline towards the weight that is at the greater distance.
2. If, when weights at certain distances are in equilibrium, something is added to one of the weights, they are not in equilibrium, but incline towards the weight to which the addition was made [4] (p. 189).

The postulates are not self-evident, and it is more simple to deny their opposites rather than affirming them, leading to the use of the reductio ad absurdum, rarely practiced by more conservative mathematicians. In what follows, Archimedes' proof of his first proposition is reported to show his typical way of reduction to absurdity.

Weights which balance at equal distances are equal. For if they are unequal, take away from the greater the difference between the two. The remainder will

not then balance [supposition 3], which is absurd [supposition 1]. Therefore the weights cannot be unequal [4] (p. 190).

Natural philosophy had a larger view; its purpose was to find the whys of natural phenomena, including those associated with living beings. Though it did not deal with the balance, nor with the lift of bodies, it devoted much investigation to the motion of bodies, a theme that is a relevant part of modern mechanics.

Aristotle addressed this topic as well, by considering motion as a process of transition from power to act. He distinguished between natural motion, that of falling heavy bodies, and violent motion, that of a projectile. In order to give an idea of the type of reasoning developed in his natural philosophy on motion, a passage is reported related to Aristotle's explanation of natural motion, in which quantitative aspects are also formulated.

Further, the truth of what we assert is plain from the following considerations. We see the same weight or body moving faster than another for two reasons, either because there is a difference in what it moves through, as between water, air, and earth, or because, other things being equal, the moving body differs from the other owing to excess of weight or of lightness. Now the medium causes a difference because it impedes the moving thing, most of all if it is moving in the opposite direction, but in a secondary degree even if it is at rest; and especially a medium that is not easily divided, that is a medium that is somewhat dense. A, then, will move through B in time G, and through D, which is thinner, in time E (if the length of B is equal to D), in proportion to the density of the hindering body. For let B be water and D air; then by so much as air is thinner and more incorporeal than water, A will move through D faster than through B. Let the speed have the same ratio to the speed, then, that air has to water. Then if air is twice as thin, the body will traverse B in twice the time that it does D, and the time G will be twice the time E [5] IV, 8, 215b.

Notice the difference in mathematical rigor with respect to Archimedes' treatise; here, there are relations that seem to be proportions between quantities, which, however, his contemporaries did not know how to measure.

## 3. New Sciences and Philosophies

In the Middle Ages, the writings of Archimedes were substantially unknown, and mechanics, always understood as a discipline that dealt with the lifting of weights, had its own development, reaching its peak with Jordanus Nemorarius in the 13th Century. Nemorarius' treatise dealing with mechanics, *De ratione ponderis* (A rational on weights), reports a quite formalized and rigorous theory, yet far from the Archimedean standards. However, for the first time in the history of mechanics, it presents a correct solution of the inclined plane by a new method, which can be seen as an embryonic form of the principle of virtual works [6] (Chapter 4.) Natural philosophy of the time followed the forms given to it by Aristotle and was interested in the causes of natural and violent motion, mainly from a qualitative point of view.

From the Renaissance onwards, Archimedean mechanics was rediscovered and went towards a synthesis with the mechanics of the Middle Ages. Aristotelian natural philosophy had entered into crisis, and its laws of motion were considered inappropriate: indeed, technological developments, in particular the introduction of artillery, made it necessary to study motion quantitatively. Mathematicians, with Niccolò Tartaglia in the forefront, began to invade the field of philosophers.

Galileo Galilei entered this groove and in about 1590 wrote *Le mecaniche* (Mechanics) [7], a treatise in which he managed to reduce the treatment of the inclined plane to that of the lever. When compared to other treatises of mixed mathematics, *Le mecaniche* includes much more of the elements of natural philosophy, concerning gravity and the actions of forces, connecting to the *Mechanicorum liber* (A book on mechanics) of

Guidobaldo dal Monte [8]. Dal Monte stressed the need of physics for mechanics as a compromise between Archimedes and Aristotle.

> Thus there are found some keen mathematicians of our time who assert that mechanics may be considered either mathematically, [removed from physical considerations], or else physically. As if, at any time, mechanics could be considered apart from either geometrical or actual motion! Surely when this distinction is made, it seems to me (to deal gently with them) that all they accomplish by putting themselves forth alternately as physicists and as mathematicians is simply that they fall between stools, as the saying goes. For mechanics can no longer be called mechanics when it is abstracted and separated from machines [8]. (Preface. Translation into English in [9]).

Dal Monte's call to physics concerned, for instance, the fact that the arms of a balance are endowed with weight; the introduction of the center of gravity as the point at which all the weight of a body is concentrated; the reference to friction; and the assumption that the fulcrum of the balance is a physical body. Dal Monte's approach, although still classifiable as Archimedean, was different in some respects from that of Archimedes himself.

Galileo dealt with another topic, that is the natural downward motion of heavy bodies. He considered the theory dealing with these problems as a new form of mixed mathematics, or a new science, and the treatise in which he reported it was published under the title *Discorsi e dimostrazioni matematiche sopra due nuove scienze* (Discourses and mathematical demonstrations concerning two new sciences) in 1638. One of the two new sciences was precisely the one that studied the fall of bodies; the other dealt with the resistance of bodies to fracture. Unlike *Le mecaniche*, in the *Discorsi*, there were many themes of traditional natural philosophy, which, in any case, were developed with a new spirit, in part outside the dominant Aristotelian conceptions of that time. The *Discorsi* is a text with many discussions and few formulas or theorems, a mixed science very much shifted towards traditional natural philosophy.

After Galileo, other mathematicians tackled typical themes of natural philosophy, as, for example, Alfonso Borelli. In his *Theoricae mediceorum planetarum ex causis physicis deducta* (Theory of planets deduced from physical causes) of 1666, he dealt with physical astronomy, introducing elements of mathematics; in this way, he broke the scheme of astronomy as divided into two parts, one mathematical (treated as a mixed science) and one physical (object of natural philosophy), and placed himself in the wake of Kepler's stripe. Indeed, Kepler started his *Paralipomena* (Appendices) of 1604 with a statement that clearly professed he was entering the field of natural philosophy.

> Albeit that since, for the time being, we here verge away from Geometry to a physical consideration, our discussion will accordingly be somewhat freer, and not everywhere assisted by diagrams and letters or bound by the chains of proofs, but, looser in its conjectures, will pursue a certain freedom in philosophizing. Despite this, I shall exert myself, if it can be done, to see that even this part be divided into propositions [10] (p. 5. Translation in [11]).

Huygens also dealt with typical themes of natural philosophy, although the figure of the mathematician was dominant in him; in any case, his considerations on the ontology of space are very interesting. On the other hand, Descartes, apart from sporadic contributions, placed himself outside the new developments of mixed mathematics and was a philosopher of nature who used mathematical concepts rather than a mathematician, who drew ideas from natural philosophy. He was one of the main forerunners of a profound change in natural philosophy, promoting what became known as *mechanical philosophy*, a new form of knowledge that had still the purpose of explaining phenomena at a qualitative level, but used only efficient causes—leaving aside formal and final causes—with their root in the motion of hard particles, which the material world was assumed to be founded by.

An important milestone for the birth of the new mechanics, which in some way constituted an official baptism, was the publication of John Wallis' treatise *Mechanica sive*

*de motu* (Mechanics, or of motion) of 1669–1671 [12]. He presented his main contribution to the solution of the problem of the motion of bodies in this book (800 pages). Wallis' treatise was the last great one of pre-Newtonian mechanics, one that, for the first time, used the word "mechanics" in the title to cover what is now called mechanics as well: "I call mechanics the geometry of motions". The *Mechanica sive de motu* was read widely, almost for sure by Newton himself; it was one of the first texts of mathematical physics in the modern sense, where writing equations is sometimes more important than understanding the meaning of the discussion in a perfect way.

## 4. Natural Philosophy of Newton and Afterwards

The most important turning point in the relation of mechanics with natural philosophy came in 1687, with the publication of Newton's *Philosophiæ naturalis principia mathematica*. This view in no way implies that there were no other scientists in the 17th and 18th Century who contributed to developing mechanics and emancipating it from natural philosophy. However, for the sake of space, an outlook on Newton is fundamental and sufficient to fix the ideas.

The story of the choice of the title of Newton's treatise is important to understand how Newton saw the relations among mathematics, mechanics, and natural philosophy. In June 1686, Newton communicated to his friend Edmund Halley (c. 1656–1743) his intention to suppress the third book of the *Principia*, dealing with the solar system, by making the following comment (ancient English retained):

> The Proof you sent me I like very well. I designed ye whole to consist of three books, the second was finished last summer being short & only wants transcribing & drawing the cuts fairly. Some new Propositions I have since thought on wch I can as well let alone. The third wants ye Theory of Comets. In Autumn last I spent two months in calculations to no purpose for want of a good method, wch made me afterwards return to ye first Book & enlarge it wth divers Propositions some relating to Comets others to other things found out last Winter. *The third I now designe to suppress* [emphasis added by us]. Philosophy is such an impertinently litigious Lady that a man had as good be engaged in Law suits as have to do with her. I found it so formerly & now I no sooner come near her again but she gives me warning. The two first books without the third will not so well beare ye title of *Philosophiæ naturalis Principia Mathematica* [Mathematical principles of natural philosophy] & therefore I had altered it to this *De motu corporum libri duo* [Two books on the motion of bodies]: but upon second thoughts I retain ye former title. Twill help ye sale of ye book wc I ought not to diminish now tis yours. The Articles are wth ye largest to be called by that name. If you please you may change ye word to *sections*, thô it be not material. In ye first page I have struck out ye words *uti posthac docebitur* [will be taught to use from now on] as referring to ye third book [13] (vol. 2, p. 437. Newton to Halley 26 June 1686).

From this excerpt, it is apparent that Newton believed that the first two volumes of his treatise were too mathematical to qualify them to deal with natural philosophy. In any case, he thought it better to keep the locution "natural philosophy" in the title, not because he meant it to actually be a treatise on this discipline, but rather for promotional reasons. Fortunately, thanks to Halley's intercession, certified by the following quote, Newton's original idea of three volumes remained intact (ancient English retained):

> Sr I must now again beg you, not to let your resentments run so high, as to deprive us of your third book, wherin the application of your Mathematicall doctrine to the Theory of Comets, and severall curious Experiments, which, as I guess by what you write, ought to compose it, will undoubtedly render it acceptable to those that will call themselves philosophers [13] (vol. 2, p. 443. Halley to Newton 29 June 1686).

This quotation makes it clear that mathematicians subsumed physical astronomy (a very relevant part of natural philosophy) under mechanics, thus completing the path started by Kepler.

A modern reader might think that non-mathematical philosophers could not even understand the third book, which of course presents mathematical difficulties for readers that are not well trained in mathematics. Moreover, the same reader, accustomed to the concise treatises of classical mechanics, would give a slightly different opinion from that of Newton. It is true that the *Principia* strongly relies on mathematics (it could even be said differential geometry), but in the treatise, important themes of natural philosophy are also found. Indeed, right from the beginning, issues about the ontology of space and time, which was an important subject of philosophy from the Middle Ages to Newton's time, are discussed. Furthermore, the concept of force, gravity in particular, is the subject of both ontological and epistemological considerations, even if in the end, Newton came out with a bit of a positivistic approach. He was more explicit in the *Querries* of his *Opticks*, which, in line with the tradition of the Middle Ages, was mainly of an experimental nature and more in accord with traditional natural philosophy. In the final scholium of the Principia, starting with the second edition of 1713, Newton introduced the role of divinity as the primary cause of any physical phenomena; this, too, is a theme of the natural philosophy of Christian Europe.

The treatises on mechanics after Newton keep the same cut somehow; going in big steps, consider Leonhard Euler's *Mechanica sive motus scientia analytice exposita* (Mechanics, or science of motion exposed analytically), published in 1736. Here, an abrupt change from the point of view of the mathematical treatment is found, since the main tool is no longer differential geometry, but rather, infinitesimal calculus. Euler was never explicit about his epistemology regarding mechanics, but only about the role of mathematics in mechanics, since he declared to prefer the analytic treatment of calculus over the geometric one. To Euler, only infinitesimal calculus allows for a systematic approach to all problems, while geometry requires an ad hoc treatment for each problem.

> Being engaged in this business, not only have I fallen upon many questions not to be found in previous tracts, to which I have been happy to provide solutions, but also I have increased our knowledge of the science by providing it with many unusual methods, by which it must be admitted that both mechanics and analysis are evidently augmented more than just a little (Preface. Translation into English by I. Bruce).

Euler's *Mechanica*, along with other writings of his, contains important insights into natural philosophy, in particular in-depth discussions on the ontology of space and force. d'Alembert had a quite different approach: he limited the considerations of natural philosophy to the introduction of his *Traité de dynamique* (A treatise of dynamics) of 1743, in which he conceived of mechanics as a particular branch of mathematics, that is a purely rational science, contrasting other scholars of the time, such as Daniel Bernoulli, who treated mechanics as an empirical science (see [14]).

Mechanics with d'Alembert and Euler had become an algebraic theory; however, it still remained geometric in some parts that relied on the concept of the vector (this is a modern term). Such a concept can have no reference to classical geometry currently, but it could not be so in the 18th Century. Lagrange took an important and large step towards a complete algebrization of mechanics in his *Méchanique analitique* of 1788. While the treatise probably added little to the theory of mechanics—indeed, it mainly collected most of Lagrange's previously obtained results—it was completely new for its logical–epistemological conception and its way of exposition: rational mechanics became analytical. Moreover, there was no room for concepts of traditional natural philosophy, in particular no references to any divinity.

Lagrange in the preface of his masterwork declared mechanics a branch of analysis, whose principles were given by general formulas. They were assumed as given, with no

interest in their derivation; from this point of view, the *Méchanique analitique* should be considered one of the first modern texts in modern mathematical physics.

Though well known and mentioned in the literature, the whole preface of Lagrange's treatise is quoted, given its shortness.

> I propose to condense the theory of this science and the method of solving the related problems to general formulas whose simple application produces all the necessary equations for the solution of each problem. I hope that my presentation achieves this purpose and leaves nothing lacking. In addition, this work will have another use. The various principles presently available will be assembled and presented from a single point of view in order to facilitate the solution of the problems of mechanics. [...] No figures will be found in this work. The methods I present require neither constructions nor geometrical or mechanical arguments, but solely algebraic operations subject to a regular and uniform procedure. *Those who appreciate mathematical analysis will see with pleasure mechanics becoming a new branch of it* [emphasis added by us] and hence, will recognize that I have enlarged its domain [15] (Preface. English translation in [16]).

## 5. Natural Philosophy and the New Sciences Relying on Mechanics in the 19th Century

With Lagrange, mechanics completely detached from traditional natural philosophy. In any case, natural philosophy in its "classical" meaning practically did not exist anymore; it had been replaced by *physics*, as far as it concerns inert matter, and by *natural history*, as far as it concerns living beings. It must be said, however, that the locutions "natural philosophy" and "physics" were used synonymously throughout the 19th Century, especially in English-speaking countries. Moreover, some treatises, especially those of non-specialized writers, remained anchored in some aspects to ancient natural philosophy with the use of formal and final causes, divinity, and certain forms of spiritualism. This approach was justified, at least until the turn of the 19th Century: indeed, the physics of the time dealt with electricity, thermometry, and light propagation (to say a few), besides mechanics. Though having an experimental approach, physicists used largely qualitative explanations, where the search for causes, not only efficient ones, was frequent, and that was typical of traditional natural philosophy. In the 19th Century, physics had become a fully quantitative science, with the aim of formulating physical laws by means of mathematical functions; moreover, it became strongly oriented toward applications following the turmoils of the French revolution, on the one hand, and the continuous asking for technological applications required by the first industrial revolution, on the other hand. The new philosophy of nature, physics, began to be thought suitable for everyday life applications and, thus, necessary for the education of young pupils of both genders, which represented both an ethic and social revolution [17,18].

The last decades of the 18th Century saw the attempt to use the concepts and categories of mechanics to pose the bases to interpret the phenomena investigated by the new branches of physics of the time, that is electricity and heat. Thus, new natural philosophy faced the task of providing a mechanical explanation of both electric force (Coulomb's experiments on its measurement by a torsion balance are well known) and of heat exchange between bodies, from which it was apparent that mechanical power can be extracted. Indeed, seeing electric charge as a volumic quantity similar to mass and electric force as a body interaction at a distance reminds us of Newton's interpretation of gravitation; an electric current was considered as the simple flow of an immaterial electric fluid. Analogously, the theory of heat exchange was based on the flow of another immaterial fluid called *caloric* from bodies at different heat levels (i.e., possessing different quantities of caloric). Thus, it seemed that relying on the well-established causes of mechanics could provide solid bases for the interpretation of new phenomena in the style of traditional natural philosophy.

A very interesting instance of the remainder of traditional natural philosophy in the new branches of science is represented by the connections of heat and *vis viva* (living force). A vivacious debate on the very nature of caloric and on its relation with *vis viva* involved such eminent scholars and applied scientists as P.-É.-B. Clapeyron, W. J. Maquorn Rankine, J. Prescott Joule, J. Clerk Maxwell, the young William Thomson (Lord Kelvin of Largs), and his brother, James. Indeed, according to natural philosophy, if heat transfer produces motion, then the caloric fluid must possess *vis viva*, which is transferred to the moved mechanism. However, in some transformations, there seems to be some heat that is lost, which should imply that *vis viva* can be lost. This was unacceptable for such thinkers as the Thomson brothers and Joule on a metaphysical basis; on the other hand, the German Rudolf Clausius did not link his ideas to metaphysical or religious beliefs: he came to postulate his thermodynamic principles as mere acknowledgments of the world and of its natural processes.

Strongly influenced by a spiritualist education at Glasgow, the Thomson brothers' investigations relied on natural phenomena, thence also on heat, by some divine establishment. Indeed, they were firmly convinced that the investigation of nature, i.e., natural philosophy, reduces to discovering divinity as the primary cause of natural phenomena, which are maintained by a divine creative effort. The young William Thomson, especially in his early studies on thermodynamics, supported the idea of caloric by Carnot and Clapeyron: just like a waterfall produces mechanical power, so does caloric in a temperature fall. That is, caloric acquires *vis viva*, which must be conserved, since the Thomsons' natural philosophy relied on the assumption that divinity is the only supreme being that can create or annihilate *vis viva*. This view was shared also by Joule:

> [Clapeyron] agrees with Mr. Carnot in referring power to vis viva developed by caloric contained in the vapour, in its passage from the temperature of the boiler to that of the condenser. I conceive that this theory, however ingenious, is opposed to the recognised principles of philosophy, because it leads to the conclusion that vis viva may be destroyed by an improper disposition of the apparatus: thus Mr. Clapeyron draws the inference, that "the temperature of the fire being from 1000° (C.) to 2000° (C.) higher than that of the boiler, there is an enormous loss of *vis viva* in the passage of the heat from the furnace into the boiler". Believing that the power to destroy belongs to the Creator alone, I entirely coincide with Roget and Faraday in the opinion, that any theory which, when carried out, demands the annihilation of [living] force, is necessarily erroneous [19] (pp. 382–383).

The debate lasted long into the mid-19th Century and came to an end only when it was accepted that heat and *vis viva* were mutually exchangeable, which, however, implied a strong philosophic jump: heat is not a new entity, but one of the forms in which *vis viva* can manifest itself in natural phenomena.

## 6. Coming Back to the Past

Contemporary mechanics investigates a very wide range of problems, from deformable bodies and fluids to non-standard materials, fatigue, damage, and multi-physical interactions (piezo-electricity, thermo-elasticity and thermo-plasticity, the mechanics of porous means and soft tissues, just to quote a few). In standard treatises, no elements of the traditional natural philosophy are to be found, even if some hints (in a very broad sense) can appear in the most elementary textbooks. These hints are about the definitions of space, bodies, and primary causes; yet, these definitions are devoid of technical meaning and belong to ordinary speech, thus to empirical intuition, just like Aristotle's contemporaries, who were not taught natural philosophy. Of course, one could rename "natural philosophy" every investigation about nature and its phenomena: in this case, there is natural philosophy also in contemporary mechanics—however, it is simply a matter of labels.

In specialized textbooks and treatises, mechanics is a physical–mathematical, or even purely mathematical, discipline. Theory is developed starting from some principles (axioms,

definitions, primitive terms), which are given for granted. In order to apply theory to actual situations, correspondence relations that assign precise numerical values to primitive entities (for instance, mass or force) must be provided. In some way, this is a return back to ancient Greek mechanics, Archimedes in particular. This process of formalization, started by Lagrange, received fundamental contributions especially by Hamilton and Jacobi.

In order to describe how a modern, formal theory (yet not reduced to pure formal logic language) can be presented, one can refer to the treatment of continuum mechanics by Clifford Truesdell in his *A first course in rational continuum mechanics* (second edition) of 1991.

Bodies are introduced as primitive terms, and their properties are partially implicitly defined by some axioms, attributing them a Boolean algebraic structure and of a partially ordered set; as a first step, no axioms about continuity are introduced, but rather on the divisibility of bodies. Even though Truesdell exhibits a high level of abstraction, he also relies on illustrative examples in his mathematical objects:

> To picture the relations among bodies, it may help to consider $\Omega$ as being the collection of all open sets in the Euclidean plane and to take $\vee$ as begin the sign of inclusion, $\subset$, so that the suggestive sketches often called "Venn diagram" are easy to draw [20] (p. 8).

The mass $M$ of a body is another primitive term, defined by a positive function that is additive, in the sense that if $\mathcal{B}$ and $\mathcal{C}$ are distinct bodies, then

$$M(\mathcal{B} \vee \mathcal{C}) = M(\mathcal{B}) + M(\mathcal{C})$$

Hence, it is natural and intuitive to introduce a Borel measure defined on any body $\mathcal{A}$:

$$M(\mathcal{A}) = \int_{\mathcal{A}} dm, \ \forall \mathcal{A} \in \Omega_M$$

where massy bodies are a subset $\Omega_M$ of the universe of bodies $\Omega_O$, i.e., $\Omega_M \subset \Omega_O$. In this case also, Truesdell does not leave $M$ as a completely abstract quantity, but links it to classical mechanics:

> The bodies of interest in mechanics have mass, as we say they are *massy* [20] (p. 16).

Force as well is seen as a primitive concept, but in this case, Truesdell does not immediately associate the term "force" with any known concept of classical mechanics, but simply lists a series of intuitive axioms:

> A *system of forces* on a universe $\Omega$ is an assignment of vectors in some inner-product space $\mathcal{F}$ to all pairs of separate bodies of $\Omega$ [20]: (pp. 19–20).

> **Axiom F1.** $\mathbf{f} : (\Omega \times \Omega)_0 \to \mathcal{F}$

The vector $\mathbf{f}(\mathcal{B}, \mathcal{C})$ is the force exerted by $\mathcal{C}$ on $\mathcal{B}$, with no reference to anything that can be interpreted as its point of application. Some definitions and theorems then follow the given axioms.

**Definition 1.** $\mathbf{f}(\mathcal{B}, \mathcal{B}^e)$ *is the resultant force on* $\mathcal{B}$ *from its exterior.*
$\mathbf{f}(\mathcal{B}, \mathcal{C}) = -\mathbf{f}(\mathcal{C}, \mathcal{B}), \ \forall (\mathcal{B}, \mathcal{C}) \in (\overline{\Omega} \times \overline{\Omega})$ *is a system of pairwise equilibrated forces.*
$\mathbf{f}(\mathcal{B}, \mathcal{B}^e) = \mathbf{0}, \ \forall B$ *is a system of balanced forces.*
$\mathbf{F}(\mathcal{B}, \mathcal{B}^e)$ *is the resultant torque of the forces acting on* $\mathcal{B}$ *from its exterior, defined as:*

$$\mathbf{F}(\mathcal{B}, \mathcal{B}^e)_{\mathbf{x_0}} = \int_{\mathcal{B}} (\mathbf{x} - \mathbf{x_0}) \times \mathbf{df}(\mathcal{B}, \mathbf{\mathcal{B}^e})$$

**Theorem 1.** (NOLL, GURTIN, AND WILLIAMS) *A system of forces is pairwise equilibrated if and only if the resultant force* $\mathbf{f}(\mathcal{B}, \mathcal{B}^e)$*, regarded as a function of* $\mathcal{B}$*, is additive on the separate bodies of* $\overline{\Omega}$ [20]: *(p. 21).*

The theorem thus requires that the following relation holds:

$$\mathbf{f}[\mathcal{B} \vee \mathcal{C}, (\mathcal{B} \vee \mathcal{C})^e] = \mathbf{f}(\mathcal{B}, \mathcal{B}^e) + \mathbf{f}(\mathcal{C}, \mathcal{C}^e)$$

Then, Truesdell advances two axioms, called the axioms of inertia, which, according to the identification of a mass-point (a body-point according to Truesdell's nomenclature) with a body, correspond to the first two laws of motion (on the other hand, "action and reaction" is somehow a consequence and, thus, included in the definition of pairwise equilibrated forces [20] (p. 22). To this aim, Truesdell makes a distinction between observable bodies, belonging to a set $\Sigma^e$, and non-observable bodies, belonging to another set $\Sigma^e$, external with respect to $\Sigma$ (with some abstraction, one can think of the so-called "fixed stars").

**Axiom I1** . There is a frame such that if $\mathbf{m}(\mathcal{B}, \varnothing)$ is constant over an open interval of time, then in that interval $\mathbf{f}(\mathcal{B}, \Sigma^e) = \mathbf{0}$, and conversely [20] (p. 65).

Here, $\mathbf{m}$ is the linear momentum of the body in motion $\varnothing$, a well-defined quantity over the mass of the body; the frame in which Axiom I1 holds is called an *inertial frame*. The axiom is formally equivalent to the corresponding one by Newton, but it refers to a body and not to a body-point.

Based on the fact that one ignores everything in $\Sigma^e$, and thus $\mathbf{f}(\mathcal{B}, \Sigma^e)$ shall depend only on $\mathcal{B}$ and on its motion, Truesdell introduces another axiom, resembling Newton's second law [20]: (p. 68).

**Axiom I2** . NEWTON, EULER, AND OTHERS. In an inertial frame

$$\mathbf{f}(\mathcal{B}, \Sigma^e) = -\dot{\mathbf{m}}(\mathcal{B}, \varnothing)$$

**7. Final Remarks**

In ancient times, the term "mechanics" was somehow restricted to the investigation of specific problems (such as the lifting of weights) and to the actual means of solving them (i.e., it had to do with "engineering"). On the other hand, the locution "natural philosophy" meant the search for an explanation of the relations of causes and effects in the events of the world—thence, it comprised the study of the motion of bodies, as well as astronomy, botanics, zoology, meteorology, and so on. A change of attitude is conventionally attributed to Galileo, who started to describe phenomena by mathematics and the introduction of time as a quantitative variable instead of trying to describe cause–effect relations. A definitive mathematization of natural philosophy is due to Newton, who, in his masterpiece, claimed to discuss natural philosophy by mathematical tools. After him, natural philosophy became what is now called physics, strongly relying on a mathematical language and detaching from tradition; only in English-speaking countries, the two locutions, "natural philosophy" and "physics", remained synonyms throughout the 19th Century. With Lagrange, mechanics became an analytical discipline where there was no room for the concepts of traditional natural philosophy, in particular no mention of any divinity. Remarkable exceptions are the discussions on heat and *vis viva* that arose in the 19th Century to account for thermodynamic transformations and the possibility to convert mechanical work into heat and vice versa, where use was made of metaphysical categories. Another exception is represented by the German current of *Naturphilosophie*, which sought to go back to the original attempt to explain natural phenomena per causas, using a not well-defined concept of force as a basic idea. Mechanics, in the sense of the investigation of the motion of bodies, which thus constituted the foundations of ancient natural philosophy, went into crisis at the dawn of the 20th Century, with quantum and relativistic mechanics. Currently, mechanics is labeled as "classical" and has become a

formalized, axiomatized physical mathematical discipline: in some sense, it has gone back to its "rational" origins as established by Aristotle and Euclid.

**Author Contributions:** D.C. and G.R. contributed equally to this manuscript. All authors have read and agreed to the published version of the manuscript.

**Funding:** This work has received no external funding.

**Institutional Review Board Statement:** Not applicable.

**Informed Consent Statement:** Not applicable.

**Data Availability Statement:** No data presented.

**Conflicts of Interest:** The authors declare no conflict of interest.

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
