# Peer review of "Mechanics and Natural Philosophy in History"

_encyclopedia, doi:10.3390/encyclopedia2030089_

Round 1
Reviewer 1 Report
The entry is missing several important and necessary elements:
- Natural philosophy had multiple meanings both before and after Galileo. Writing that natural philosophy was about the identification of "the physical causes of all natural effects" is an anachronism (physical causes?) and oversimplification. Nothing about the concept of "nature/natural" in Aristotelian and Scholastic thought.
- Involvement of mathematics was much more complicated and at least postulated before Galileo (Roger Bacon).
- Missing the fundamental idea of "saving appearances" making the distinction between natural philosophy and mechanics understood as creating mathematical models consistent with observed phenomena.
- Nothing about the question about the status of Copernican description of the world (saving appearances or natural philosophy?)
- The authors focus exclusively on Newton, as if Francis Bacon, Descartes, and others did not exist.
- Not even one word about Descartes' mechanics and its three principles in "Le Monde" (1664). Nothing about the crucial difference between his mechanics exclusively with interaction on contact and Newton's action on the distance. Nothing about Newton's absolute space in contrast to Leibniz's "De analysis situ" and long lasting discussions of the status of the space.
- Newton used Euclid's Elements as a pattern for his Principia. No comment about it. The claim that Newton thought about the two volumes of his treatise as "too mathematical" is strange. If Elements were good as a pattern for Hobbes to axiomatize political science, who would object to the use of mathematics in Principia.
- There is nothing about the transformation of mechanics by Lagrange and Hamilton based on the principle of minimal action.
- Nothing about conservation laws.
- Nothing about symmetry (Noether's theorem).
- Nothing about Laplace and mechanical determinism.
- Nothing about physicalism and reductionism.
- Nothing about the present attempts to revitalize natural philosophy without reductionism.
The list of missing key events and ideas is too long to provide more extensive comments on each of them. However, they are of such fundamental importance that they can be easily identified.
Most of the text is without mathematical technicalities. However, the marginal for the subject Truesdell's approach to the continuum mechanics is presented with some mathematical details. What for?
In general, the entry is written with a complete lack of judgment regarding what is important and what is not. I doubt that anyone can learn anything about the subject from this text.
Author Response
The authors wish to thank the anonymous Reviewer for his/her attention in reading our submitted manuscript and for his/her criticisms, which are numerous and thus show interest in the subject we dealt with.
We believe that the points raised by the Reviewer have a motivation, which we will discuss below. However, we feel compelled to stress that our manuscript is, by request of the Editor, as an entry paper, it cannot ever have the depth and the details of a monograph. Owing to this, a lot of possible additional connections of mathematics with mechanics, and/or with other scholars of the past were neglected. This was done not because those connections are meaningless but simply for sake of space and with the aim of keeping along the same path, however narrow and maybe subject to criticism, as the Reviewer did. Thus, to begin with, we added a couple of lines in the Definition to explain the narrowness of our presentation.
In the following we try to reply to some of the objections raised by the Reviewer, by interposing our answers to the Reviewer’s remarks:
****Natural philosophy had multiple meanings both before and after Galileo. Writing that natural philosophy was about the identification of "the physical causes of all natural effects" is an anachronism (physical causes?) and oversimplification. Nothing about the concept of "nature/natural" in Aristotelian and Scholastic thought.
The Reviewer is right in remarking that the locution ‘natural philosophy’ has many meanings and interpretations. As we said right from the beginning, we spent only a few words to describe it for the sake of space. The anachronism highlighted by the Reviewer, however, is not ours, but directly quoted from a well-known monograph by Edward Grant, one of the greatest historians of Middle Ages. Thus, we believe it is not our task to modify the original source. In any case, in the revised version we stressed that we are oversimplifying our definitions.
****Involvement of mathematics was much more complicated and at least postulated before Galileo (Roger Bacon).
We agree, and, indeed, in no place in the manuscript we stated otherwise; as usual, we limited ourselves to a sketch, since we could not enter the details about the root of the involvement of mathematics with mechanics, dating back at least to ancient Greece, much before Galileo. However, since it is a reasonable remark, we stressed this point and quoted another short contribution of ours, without pretending to give details on the subject, which would deserve an extensive treatment on its own.
*****Missing the fundamental idea of "saving appearances" making the distinction between natural philosophy and mechanics understood as creating mathematical models consistent with observed phenomena. Nothing about the question about the status of Copernican description of the world (saving appearances or natural philosophy?)
The Reviewer points out a problem that is known to the authors, who decided to avoid its mention and discussion in the submitted manuscript due to the following reasons:
- Sake of space.
- It was problem concerning mathematics, rather than mechanics, and natural philosophy.
- It was of interest for some philosophers more than for mathematicians, who – apart from some astronomers of Middle Ages and Renaissance – believed that their assertions were about the true world, and not only to save phenomena (this was also Copernicus’s idea, pace Osiander).
Thus, though the Reviewer points out a meaningful problem in the history of science, we feel it is out of the limited scope of this entry and did not modify anything in the revised version.
*****The authors focus exclusively on Newton, as if Francis Bacon, Descartes, and others did not exist.
Not even one word about Descartes' mechanics and its three principles in "Le Monde" (1664). Nothing about the crucial difference between his mechanics exclusively with interaction on contact and Newton's action on the distance. Nothing about Newton's absolute space in contrast to Leibniz's "De analysis situ" and long lasting discussions of the status of the space. Newton used Euclid's Elements as a pattern for his Principia. No comment about it.
The Reviewer recalls fundamental scientists in mechanics and rises a point about our stress on Newton. The great role we give to Newton in exposing our view of the relation of natural philosophy and mechanics is due to the following reasons:
- For the sake of space: we ignored Descartes and Leibniz as well as other great scientists such as Huygens, Torricelli, Hooke, Roberval, and so on.
- Differently from many British historians, we believe that Bacon’s role in mechanics is not so relevant.
However, since the Reviewer might have a point in recalling other important scientists, we highlighted in the revised text that our presentation is limited, and the interested reader shall deepen the knowledge of these other scientists.
*****The claim that Newton thought about the two volumes of his treatise as "too mathematical" is strange. If Elements were good as a pattern for Hobbes to axiomatize political science, who would object to the use of mathematics in Principia.
The quote about an excessively mathematical treatment in the Principia is Newton’s, not ours: as above, we do not wish to change the original sources.
*****There is nothing about the transformation of mechanics by Lagrange and Hamilton based on the principle of minimal action.
The Reviewer might have a good point if we were to produce a more extensive contribution; however, the authors had to cut things, and in this regard believe that these aspects were to be avoided for the sake of space. One of the authors discussed the problem elsewhere, as it is quoted in the text, see: The problem of motion of bodies, Springer 2014. For this reason, we added only a very short hint in the revised manuscript.
*****Nothing about conservation laws.
Nothing about symmetry (Noether's theorem).
Nothing about Laplace and mechanical determinism.
Nothing about physicalism and reductionism.
Nothing about the present attempts to revitalize natural philosophy without reductionism.
The list of missing key events and ideas is too long to provide more extensive comments on each of them. However, they are of such fundamental importance that they can be easily identified.
The Reviewer points out a series of subjects that are not central to the entry from our point of view and are thus ignored for the sake of space. Since they are, as the Reviewer himself remarks, numerous and not trivial, we decided not to mention them also in the revised version.
*****Most of the text is without mathematical technicalities. However, the marginal for the subject Truesdell's approach to the continuum mechanics is presented with some mathematical details. What for?
The referee is right on this point. Truesdell mathematical treatment should have been counterbalanced by a similar one due to Archimedes, to which it is explicitly compared. The reason of the greater space given to Truesdell than to Archimedes is still due to the sake of space, and because Archimedes is more known than Truesdell to Historians, which allows to resume his approach in short.
Reviewer 2 Report
The manuscript "Mechanics and Natural Philosophy in History" authored by Danilo Capecchi and Giuseppe Ruta provides a historical tour on those two concepts, fundamental in the emergence of modern science. Given the relatively short extension, the history becomes rather crammed, although the essential periods are well covered. Given that it is addressed to an engineering encyclopedia, perhaps some brief reference to the role of military (& civilian) engineers in the classical period of mechanics helenistic and Roman could be in order. In Archimedes, Vitrubius, Galileo and others the military engineering is not too far. But this suggestion is not a forceful request --only a suggestion.
Author Response
The authors wish to thank the Reviewer for his/her suggestion to refer to the engineering world, which is for sure interesting. However, this entry is a short communication and we focussed on mechanics as a `pure' science and his relations with natural philosophy as `pure' rational investigation. Thus, we decided to neglect details related to the meaning given to the term `mechanics' in the technical environments. In any case, since the Reviewer's suggestion is quite meaningful, we inserted some lines in the first section where we better specify the outlook of the contribution and the neglecting of the relations with technology.
Reviewer 3 Report
I think the manuscript is a fair and objective overview of the subject. I agree that the connection between natural philosophy and mechanics is with to be investigated and I think the authors gives a possible point of view. This is only one possible point of view but to investigate all the aspects I think that we need a book and not an article. So my opinion is that the manuscript is worth to be published.
Author Response
The authors wish to thank the anonymous Reviewer for his/her effort to read and evaluate this short communication; we understand that in so few pages it is rather complicated to provide details in full on such an extensive subject b ut we are happy that the Reviewer clearly had this point in mind. Many thanks for your kind evaluation.
Reviewer 4 Report
The paper is well documented and exciting. Historical elements pursued in their development are presented attractively and exhaustively, an excellent introduction to the context.
Author Response
The authors wish to thank the anonymous Reviewer for his/her effort to read and evaluate this short communication; we understand that in so few pages it is rather complicated to provide details in full on such an extensive subject but we are happy that the Reviewer clearly had this point in mind. Many thanks for your kind evaluation.
Round 2
Reviewer 1 Report
I don't see any improvement in the revised version of the submission. The authors focused more on making the text criticism-free than on the actual improvement in a similar way. The clearest example of such an attempt can be found in the revised abstract:
"Since we are not concerned with the summation of the histories of natural philosophy and mechanics, but only with their interrelations, this makes a detailed description of the two disciplines unnecessary."
Basically, all my objections from the original (critical) review apply to the revised version. Actually, the revision made the manuscript worse in many respects. A few additional comments.
The authors responded that they did not write anything about Francis Bacon because he did not contribute to mechanics. Of course, he didn't! But he promoted the view of the primary role of inductive reasoning over deductive one. Without mentioning this relationship the entry is clearly deficient.
The authors ignored the central issue of the role of the variational principles of mechanics which stimulated the discussion on the relationship between mathematics and natural philosophy in the 19th and early 20th centuries with participants such as Mach, Hilbert, Klein, Einstein, Pauli.
The dismissal of the central issue of the early expression of the relationship between mechanics and natural philosophy in the controversies over "saving appearances" shows that the authors really want to close their eyes not to see the most important aspects of the subject of the entry. No, when you close your eyes not to see the central themes of the subject, they are not going away.
In short, why the submission is bad and should not be published:
1) The lack of focus: an entry in an encyclopedia requires making choices, but the choices made by the authors seem random, and what was chosen is marginal.
2) The lack of consistency to whom the entry is addressed. For a wide audience, avoidance of mathematical formalism is a wise choice. However, what is the rationale to include the presentation of Truesdell's formalism? Especially, considering that from the historical perspective, his contribution is marginal.
The submission is a bad academic writing and should not be published.
Author Response
We are sorry that our responses did not meet the Reviewer's points. Actually, we believe we do not share the same outlook on the subject, since the Reviewer points out reasonable criticism for an thorough, much more extensive, contribution with respect to the introductory one we provided, which was aimed at a very limited sector of an enormously vast subject. Many thanks in any case for having read and commented our contribution.